# Association between Prescribing and Intoxication Rates for Selected Psychotropic Drugs: A Longitudinal Observational Study

**DOI:** 10.3390/ph17010143

**Published:** 2024-01-22

**Authors:** Matej Dobravc Verbič, Iztok Grabnar, Miran Brvar

**Affiliations:** 1Centre for Clinical Toxicology and Pharmacology, University Medical Centre Ljubljana, 1000 Ljubljana, Slovenia; miran.brvar@kclj.si; 2The Department of Biopharmaceutics and Pharmacokinetics, Faculty of Pharmacy, University of Ljubljana, 1000 Ljubljana, Slovenia; 3Centre for Clinical Physiology, Faculty of Medicine, University of Ljubljana, 1000 Ljubljana, Slovenia

**Keywords:** intoxication, psychotropic drugs, prescribing, correlation, risk assessment

## Abstract

Psychotropic prescription drugs are commonly involved in intoxication events. The study’s aim was to determine a comparative risk for intoxication in relation to prescribing rates for individual drugs. This was a nationwide observational study in Slovenian adults between 2015 and 2021. Intoxication events with psychotropic drugs were collected from the National Register of intoxications. Dispensing data, expressed in defined daily doses, were provided by the Health Insurance Institute of Slovenia. Intoxication/prescribing ratio values were calculated. The correlation between trends in prescribing and intoxication rates was assessed using the Pearson correlation coefficient. In total, 2640 intoxication cases with psychotropic prescription drugs were registered. Anxiolytics and antipsychotics were the predominant groups. Midazolam, chlormethiazole, clonazepam, sulpiride, and quetiapine demonstrated the highest risk of intoxication, while all antidepressants had a risk several times lower. The best trend correlation was found for the prescribing period of 2 years before the intoxication events. An increase of 1,000,000 defined daily doses prescribed resulted in an increase of fifty intoxication events for antipsychotics, twenty events for antiepileptics, and five events for antidepressants. Intoxication/prescribing ratio calculation allowed for a quantitative comparison of the risk for intoxication in relation to the prescribing rates for psychotropic drugs, providing additional understanding of their toxicoepidemiology.

## 1. Introduction

Psychotropic prescription drugs, together with analgesics, represent 70–80% of registered intoxication cases due to medication [1,2,3,4]. They also account for up to 80% of intentional drug overdoses due to prescription medication and are an increasing cause of hospitalisation in this context [1,5,6,7,8]. Suicidal attempts occur in more than 75% of the intoxication cases with benzodiazepine receptor agonists and antidepressants [9]. Patients taking psychotropic medication commonly suffer from one or several mental illnesses, e.g., depression or anxiety, which by itself may be one of the reasons for intentional overdose [1,3,5]. Almost 80% of patients with medication overdose have previously consulted a psychiatrist [9]. Psychotropic prescription drugs are generally required for optimal treatment of mental illness [10]. At the same time, they require caution and vigilance to prevent the risk of intentional or unintentional intoxication events. Previous studies have shown that patients primarily use their own medication in self-harm acts [1,11].

The comparative risk for intoxication with different psychotropic medication is typically determined by the absolute number of intoxication events for each drug [2,3,8,10,11,12,13,14,15]. Previous studies have also compared the number of intoxication events with the prescribing rates of psychotropic drugs [1,2,5,6,12,13,16,17,18]. Suicide or suicide attempt rates have been compared to psychotropic prescribing in several countries [19,20,21,22,23]. However, no direct quantitative comparison of the overall risk for intoxication with psychotropic drugs from different groups has been performed in relation to their prescribing rates.

A link between changes in prescribing patterns and intentional intoxications for psychotropic drugs over time periods has been established [1,5,6,8,11,16]. Forster et al. reported that a reduction of 1000 prescriptions for psychotropic drugs was related to 3.8 fewer cases of admission due to medication self-poisoning [17]. A similar relationship has not been observed for all agents. Increasing prescribing rates of antidepressants in Australia have not led to higher intoxication rates [10]. Furthermore, changes in intoxication rates may have occurred with a time gap from changes in prescribing patterns [12]. Improved understanding of the association between prescribing and intoxication rates can help prevent self-harm acts, including suicide attempts with psychotropic prescription drugs, in the future [5,8].

The aim of the study was to assess the risk for intoxication with psychotropic drugs at the national level by determining the association between prescribing and intoxication rates. Five drug groups were included: antipsychotics, anxiolytics, hypnotics/sedatives, antidepressants, and antiepileptics. Intoxication rates in relation to the prescribing rates were determined by calculating the intoxication/prescribing ratio (IPR) for individual drugs. Additionally, the correlation between the trends in prescribing and intoxication rates was assessed, considering a possible time gap between the prescribing period and the intoxication events.

## 2. Results

Between 2013 and 2021, the prescribing rates for central nervous system (CNS) drugs (anatomical therapeutic chemical (ATC) group N) for adult patients have increased by 18.7% (Table 1). Rates were steadily increasing for antidepressants, antipsychotics, and antiepileptics, and were decreasing for anxiolytics and hypnotics/sedatives. During this period, antidepressants represented 40.9% of the CNS drugs prescribed. They were followed by analgesics (14.6%—not further analysed in the study), anxiolytics (9.3%), antipsychotics (8.6%), antiepileptics (7.7%, including pregabalin and gabapentin), and hypnotics/sedatives (7.4%).

Between 2015 and 2021, 2640 intoxication cases with CNS prescription drugs were registered. Of these, women represented 1475 cases (55.9%) and men represented 1159 cases (43.8%); in six cases, the gender was unknown. The median age was 44 years. Relative to the number of all registered intoxication events, these rates increased by 34.6% during the 7 year period (from 26.6% in 2015 to 35.8% in 2021) (Table 2). No steady increasing or decreasing trend was observed between 2015 and 2021. 

### 2.1. Intoxication/Prescribing Ratio

Table 3 demonstrates the intoxication rates and the 7 year IPR values for the five groups of psychotropic drugs, calculated for the period of 2015–2021 (per 1000 registered intoxication events and per 100,000 DDDs). The trends in the relative intoxication rates and the IPR values for the five psychotropic groups for the period of 2015–2021 are presented in Figure 1 and Figure 2. 

For anxiolytics and antipsychotics, which represented the highest risk groups, there were no major differences between the trends for relative intoxication rates and for IPR values. On the other hand, these trends differed for antidepressants, antiepileptics, and hypnotics/sedatives. Antidepressants followed anxiolytics and antipsychotics in absolute intoxication rates, but demonstrated the lowest risk for intoxication according to IPR values. Hypnotics/sedatives had a higher risk for intoxication according to IPR calculations than antiepileptics and antidepressants. For individual agents, IPR values are presented in Table 4.

For individual psychotropic drugs, the risk for intoxication according to the IPR values differed substantially from the intoxication rates. Quetiapine, alprazolam, zolpidem, and diazepam were the most commonly involved individual drugs, while IPR for midazolam was several times higher than for any other psychotropic drug. It was followed by chlormethiazole, clonazepam, and sulpiride. Of the five drugs with the highest intoxication rates, quetiapine had the highest 7 year IPR (2.49). The IPR for zolpidem, the most commonly prescribed hypnotic, was relatively low compared to other hypnotics/sedatives, anxiolytics, and the majority of antipsychotics throughout the entire 7 year period. On the other hand, even though anxiolytics had the highest 7 year IPR as a group (Table 3), individual agents (e.g., diazepam, alprazolam, bromazepam) ranked only after several hypnotics/sedatives and antipsychotics. All antidepressants had IPR values several times lower than the highest-ranking drugs. Trends in relative intoxication rates and IPR values for the five psychotropic drugs with the highest 7 year IPR values are shown in the Appendix A (Appendix A).

### 2.2. Correlation between Trends in Prescribing and Intoxication Rates

The correlation between trends in prescribing and intoxication rates was evaluated for three prescribing periods with respect to the intoxication events. The best correlation was demonstrated for a 2 year gap, i.e., the prescribing period of 2 years before the registered year of the intoxication events (Table 5; details in Appendix A). 

The correlation was significant for antiepileptics, antipsychotics, and antidepressants (*p* = 0.001; 0.021; 0.045, respectively, for a 2 year gap). However, the slopes of the correlation plots were different for the three groups. Taking into account the average number of 1000 intoxication events per year, the increase in the prescribing rates of 1,000,000 DDDs resulted in an increase of 50 intoxication events for antipsychotics, 20 events for antiepileptics, and five events for antidepressants (Figure 3). 

The correlation fell short of statistical significance for the prescribing rates of the year of the intoxication events (i.e., no time gap) for antidepressants, whereas for antipsychotics, this was the case for a 1 year gap between the prescribing and intoxication rates.

## 3. Discussion

### 3.1. Intoxication/Prescribing Ratio 

In this longitudinal study, the risk for intoxication for individual psychotropic drugs and drug groups was determined using IPR calculations, and by associating prescribing and intoxication rates. The highest risk for intoxication was demonstrated for midazolam, chlormethiazole, clonazepam, sulpiride, and quetiapine. As psychotropic drug groups, anxiolytics and antipsychotics had the highest IPR values, while the risk for intoxication was several times lower for antidepressants. These results demonstrate that the risk for intoxication for an individual agent should not be derived from the risk for a psychotropic drug group. The most relevant differences were observed in the group of hypnotics/sedatives, which had a lower risk than anxiolytics or antipsychotics, while the individual hypnotic drugs midazolam and chlormethiazole were the highest risk agents.

Among the 15 drugs with the highest IPR values, 7 were BDZs (Table 4). In Melbourne, alprazolam and diazepam were over-represented among the BDZs in overdose patients compared to their prescribing rates [2]. In this study, the same two agents were involved in the highest number of intoxication events among BDZs. However, midazolam and clonazepam presented an even greater risk for intoxication, as their IPR values were several times higher (26.59 and 6.08, respectively, compared to 1.45 and 1.95 for alprazolam and diazepam). While safety measures were implemented in Slovenia for several BDZs in 2018 [24], additional vigilance appears to be necessary for quetiapine due to the increasing intoxication rates in the last years. Since 2019, it has become the leading cause of intoxication with a psychotropic, as well as with any type of medication. The prescribing rates of quetiapine in the last years have been increasing in many countries [25]. It has been commonly used for off-label conditions, e.g., to treat or prevent insomnia or anxiety [13,15,26,27], and has been increasingly misused and abused [27,28,29]. Similarly, intoxication rates, ambulatory visits, and fatal poisoning events related to quetiapine have increased [10,18,28,30,31]. In Victoria, a positive correlation was observed between increased prescribing and overdoses; as well, mortality due to quetiapine was observed. The rate of quetiapine overdose cases per 100,000 prescriptions increased from 37.2 to 49.3 between 2006 and 2015 [30]. Increased surveillance with the prescribing of quetiapine has already been suggested in certain countries [26,27,32]. Taking into account the calculated IPR values, closer monitoring may also be recommended for patients taking midazolam, chlormethiazole, clonazepam, sulpiride, and promazine. 

All included BDZs and the majority of antipsychotics, both commonly used as hypnotics or sedatives [33,34], had higher IPR values than any of the individual antidepressants. Trazodone, the antidepressant with the highest IPR (0.72), is also used off-label to treat insomnia [35,36]. Of the five psychotropic drug groups, antidepressants had the lowest risk for intoxication. In Flanders, prescribing rates and use in self-harm acts increased in 2008–2013 and had a positive correlation for antidepressants and antipsychotics [5]. In Australia, however, self-poisoning rates did not follow the increase in antidepressant prescribing, and intoxication rates per DDDs dropped significantly from 1995–2010. Several reasons for this were suggested. Antidepressants may be prescribed more frequently to patients with minimal risk for self-harm. Furthermore, reducing the burden of depression should lead to lower rates of self-harm acts [10]. Therefore, withholding antidepressants might lead to an increase rather than a decrease in self-harm behaviour.

There may have been additional factors influencing IPR values that were not captured in the analysis. In 2018, national restrictions were placed on the prescribing of several BDZs at higher oral doses (alprazolam 1 mg, diazepam 10 mg, midazolam 7.5 mg and 15 mg, and bromazepam 6 mg) [24,33], which had an impact on their prescribing rates. Midazolam was particularly interesting in this respect. The number of prescriptions with full coverage for midazolam dropped from more than 73,000 in 2018 to less than 14,000 in 2019. At the same time, there was a tenfold increase in the number of non-reimbursed prescriptions. Another factor influencing the IPR in the case of midazolam was the frequently reported use of the prescription drug Flormidal^®^, which is not available in Slovenia. Therefore, it had to be imported from abroad and was not included in the calculation of the DDDs. These factors partly explain the increasing IPR values for midazolam in the last years (Appendix A). However, even with the inclusion of the non-reimbursed prescriptions and the exclusion of the intoxication events involving Flormidal^®^, midazolam would have remained the drug with the highest IPR. For the other psychotropic drugs, the number of non-reimbursed compared to covered prescriptions was small, and including them in the analysis would not have had any relevant impact on the results.

DDDs may be more appropriate than the number of prescriptions or the number of units in the assessment of prescribing rates over time. However, variations for different indications or treatment goals are not taken into account with DDDs. Psychotropic drugs may be used outside of their registered indications at lower doses than recommended for their main indication. As an example, the DDD for quetiapine is set to 400 mg [37]. In practice, it is frequently used as a sedative at a dose of 25 mg or 12.5 mg [38]. In such cases, off-label prescribing obviously influenced the prescribing rates expressed in the DDDs. While off-label use may be a risk factor for intoxication events [15], our study was not designed to confirm this assumption.

Worldwide, the COVID-19 pandemic had a varying impact on the intoxication rates of psychotropic drugs. In 2020, there was an increase in suicide attempts with BDZ-receptor agonists in northern Poland [36]. Similarly, self-poisoning events with sedatives and psychotropic drugs (i.e., antidepressants, antipsychotics, and psychostimulants) increased in the first year of the COVID-19 pandemic in an English hospital [39]. In Sri Lanka, self-poisoning events due to medication decreased during the first wave of the pandemic, but increased during the second wave [40]. We did not observe any changes in intoxication rates related to the COVID-19 pandemic (years 2020–2021), although an influence cannot be excluded with certainty. No increasing trend was observed in prescribing rates for anxiolytics, hypnotics, and sedatives, while rates for antidepressants, antipsychotics, and antiepileptics during the pandemic continued to rise as before. Future studies that would include post-pandemic years could reveal patterns that were not recognised in the current study.

To reduce the number of intoxication events, the general public should be encouraged to return unused psychotropic medication to the pharmacy [2,5]. Prescribing psychotropic drugs in smaller quantities and without repeat prescriptions remains crucial as a strategy to prevent intoxication episodes [5]. A thorough risk assessment for substance use disorder before initiating a high-risk psychotropic drug, and close monitoring for misuse during treatment should be considered [27]. Interestingly, a recent US study showed that lower baseline exposure (lower prescribed daily dose) and dispensing a fewer days’ supply increased the risk for overdose with BDZs among regular users [41]. These results argue against the assumption that higher prescribing rates lead to higher intoxication rates for every drug. Future studies may further investigate the differences between individual drugs.

### 3.2. Correlation between Trends in Prescribing and Intoxication Rates

A 2 year gap between prescribing and intoxication events demonstrated the best correlation, with statistical significance for antipsychotics, antiepileptics, and antidepressants. These results suggest that single year IPR values should be followed for at least 2 or 3 consecutive years when comparing the risk for intoxication between different agents.

Crombie et al. demonstrated an obvious association between the number of prescriptions and hospital admissions due to self-poisoning with BDZs, barbiturates, antidepressants, antipsychotics, and opioids in Scotland [16]. There was no reported time gap between changes in prescribing and intoxication rates. Another study in England and Wales demonstrated a link between prescribing rates and hospital admissions due to psychotropic drug poisoning without a stated time gap [6]. On the other hand, an impact of legislative changes (restriction of the use of thioridazine) on reduced hospital admissions due to overdose in Scotland was observed only after 2–3 years. Unused medication may have been stored or stockpiled for a longer period before being used in overdose. The authors suggested that prescribing patterns should be reviewed over several years before evaluating their impact on intoxication events [12]. In Australia, intoxication and mortality rates associated with quetiapine between 2013 and 2015 continued to increase, although prescribing rates levelled off during this period [30]. Buykx et al. used a 2 year time gap when comparing prescribing rates and medication overdoses that required emergency department attendance. They noticed that on several occasions, medications used in overdose had been prescribed and discontinued some time before the event [2]. To our knowledge, there were no previous studies comparing different prescribing periods in relation to intoxication events.

No positive correlation was observed for anxiolytics and hypnotics/sedatives. The unexpected negative correlation for certain drugs (e.g., bromazepam) was likely due to a small sample size of the intoxication events for individual drugs and confounding factors, e.g., national restrictions on prescribing, possibly related to the practice of importing drugs from neighbouring countries. Tournier et al. found that in intentional overdose cases, patients commonly used antidepressants, antipsychotics, and antiepileptics as their own prescribed medications, while anxiolytics and hypnotics were used in overdoses indiscriminately [8]. Even though these findings may not be generalizable, they may provide a reasonable explanation for the absence of a positive correlation between the time of prescribing and intoxication for anxiolytics and hypnotics/sedatives in this study. 

Given the recognised correlation gap of 2 years, it is advisable to calculate a single year IPR for 2 or 3 consecutive years. An average IPR value for several years may provide a more reliable estimate for the comparison of the risk for different psychotropic drugs (or drug groups). In this study, calculation of a 7 year IPR was performed. The authors discourage adjusting the single year IPR calculations by using the prescribing rates from 2 years before the intoxication events. For individual drugs, this would reduce year to year changes in IPR values, which in themselves are informative and should be addressed when noticed. 

### 3.3. Study Limitations

Using data from the National Register of intoxications inevitably led to a selection bias. Fatal cases are generally not reported to the Toxicology Service and, therefore, are rarely registered. Less severe, medically unidentified intoxications may also be under-reported [13].

Due to evolving methods of recognition and registering the intoxication events in the National Register, the absolute number of registered intoxication events per year has increased during the analysed period. In order to avoid misinterpretations, the intoxication rates of psychotropic drugs were also presented relative to the number of all registered intoxication cases. 

Data are registered in the National Register of intoxications based on the received telephone calls and/or medical records. Patients, accompanying persons, or medical staff may report subjective or incomplete information [42]. There may have been occurrences of misclassification (registering drugs of abuse as prescription drugs) or similar errors, because the information on the intoxication events may not always have been accurate. Drug concentrations are not routinely tested to confirm intoxications. Nevertheless, previous studies for specific drugs demonstrated that patient history was generally confirmed by appropriate assays [10,43].

Prescribing rates were estimated from the dispensing data for the reimbursed outpatient prescriptions, which may not entirely reflect the prescribing patterns. On the other hand, dispensing data may better reflect the availability of a drug for a potential intoxication.

Small sample size for individual drugs (≥15 intoxication cases on average per year), a relatively short study period of 7 years, and multiple comparisons may have biased the results of the correlation analysis.

The risk for intoxication with a psychotropic drug was defined as the likelihood of an intoxication occurring and being registered in relation to prescribed DDDs. The IPR is not related to the severity and outcomes of intoxication events and does not differentiate between intentional and unintentional intoxication events. However, it has previously been recognised that in the intoxication cases reported as ‘intentional’ or ‘deliberate’, the extent to which the act had been intentional was not always clear [5].

Finally, prescribing and intoxication patterns may differ in specific subgroups of patients [44,45,46]. This analysis was performed exclusively on the general adult population, and the data were not stratified by age, gender, or disease.

## 4. Materials and Methods

### 4.1. Data Collection

This was a longitudinal study analysing intoxication events and their association with prescribing rates at the national level in Slovenia, a member country of the European Union with 2.1 million inhabitants [47]. Intentional and unintentional intoxication events with psychotropic prescription drugs in adult patients between 2015 and 2021 were included, regardless of the severity of symptoms, the outcome, or the co-ingested agents. Intoxications not involving psychotropic prescription drugs (e.g., intoxications with drugs of abuse or medication obtained from sources other than by prescription) were excluded. Anonymised data were extracted from the National Register of intoxications. 

Prescribing rates were estimated from the dispensing data for CNS drugs for adult patients provided by the Health Insurance Institute of Slovenia for the period of 2013–2021. Only outpatient prescriptions with national coverage were included. The defined daily dose (DDD), i.e., the average daily adult dose for the main indication of a specific drug for the oral route of administration, was used as a prescribing unit in the analysis [37,48]. Five ATC classification groups of psychotropic drugs were analysed: N03—antiepileptics, N05A—antipsychotics, N05B—anxiolytics, N05C—hypnotics and sedatives, and N06A—antidepressants. Other psychotropic drug groups were not included, as they represented a minor proportion of intoxication cases in Slovenia.

### 4.2. Data Analysis

The absolute intoxication rates were obtained for five psychotropic drug groups and for individual drugs as the number of intoxication events in a single year or during the 7 year period of 2015–2021. In addition, the relative intoxication rates with the corresponding 95% confidence intervals were determined to avoid potential bias due to evolving methods for the recognition and registration of the intoxication events over time. These were calculated as the number of intoxication events divided by the number of all intoxication events in the National Register for the same year or 7 year period. The intoxication/prescribing ratio (IPR) values were calculated by dividing the relative intoxication rates with the prescribing rates, expressed in DDDs. To provide the results per 1000 registered intoxication events and per 100,000 DDDs, the IPR values were multiplied by a factor of 10^8^. The IPR values were determined for the five psychotropic drug groups and for individual agents that were involved in ≥20 intoxication cases in 2015–2021. 

The correlation between the trends in prescribing and intoxication rates was assessed using the Pearson correlation coefficient. A *p*-value of <0.05 was set as the threshold for statistical significance. The prescribing rates (expressed in DDDs) were collected and compared with the intoxication rates with (a) no time gap; (b) a 1 year gap; or (c) a 2 year gap between the time series [year] of prescribing and the time series [year] of intoxication events. The correlation between intoxication and prescribing rates was calculated for the five psychotropic drug groups, and for individual agents involved in ≥15 intoxication cases on average per year. The best correlation was determined as the lowest *p*-value derived from the Pearson correlation coefficient. 

The study conformed to the bioethics and clinical research data protection legislation. The STROBE guidelines for observational studies were followed where applicable. As this was a retrospective, noninterventional analysis of data from two anonymised databases, an approval by the National Medical Ethics Committee was not required. 

## 5. Conclusions

Intoxication/prescribing ratio values differed substantially from the intoxication rates for the individual psychotropic drugs and provided an improved understanding of the risk for intoxication. Midazolam, chlormethiazole, clonazepam, sulpiride, and quetiapine were the psychotropic drugs with the highest IPR values. The best correlation between trends in prescribing and intoxication rates was obtained for a 2 year gap, with statistical significance for antipsychotics, antiepileptics, and antidepressants. The association between prescribing and intoxication events is an important factor when deciding on future measures to improve patient safety.

## Figures and Tables

**Figure 1 pharmaceuticals-17-00143-f001:**
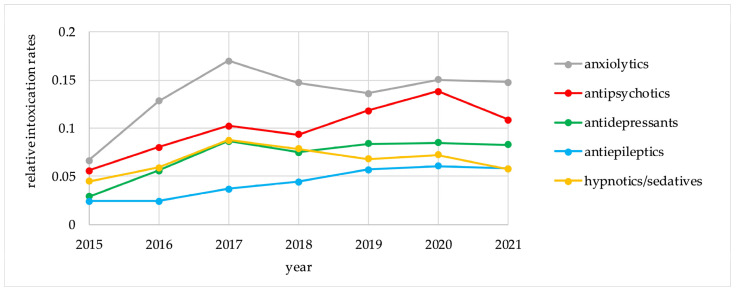
Relative intoxication rates for psychotropic drug groups, 2015–2021.

**Figure 2 pharmaceuticals-17-00143-f002:**
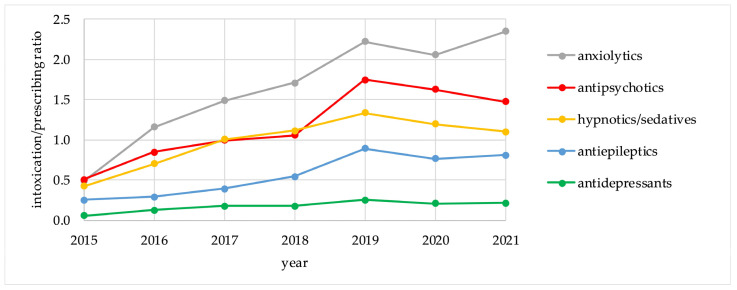
Intoxication/Prescribing ratio values for psychotropic drug groups, 2015–2021.

**Figure 3 pharmaceuticals-17-00143-f003:**
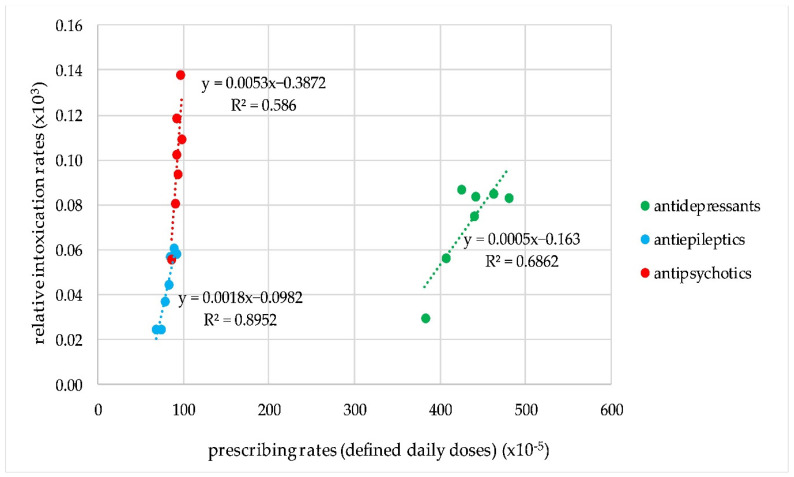
Correlation plots for the trends in prescribing and intoxication rates for antipsychotics, antiepileptics, and antidepressants.

**Table 1 pharmaceuticals-17-00143-t001:** Prescribing rates of selected psychotropic drug groups (change from 2013–2021).

Psychotropic Drug Group	Prescribing Rates		
	DDDs/100,000 Inhabitants ^b^		Change from 2013 to 2021 (%)
	2013	2021	
antidepressants	22,418	29,086	+29.7
antiepileptics	3979	5628	+41.4
antipsychotics	5026	5936	+18.1
anxiolytics	7172	4961	−30.8
hypnotics/sedatives	5577	4134	−25.9
CNS drugs ^a^	57,742	68,513	+18.7

^a^ CNS: central nervous system; ^b^ DDDs: defined daily doses.

**Table 2 pharmaceuticals-17-00143-t002:** Absolute and relative intoxication rates for central nervous system prescription drugs, years 2015–2021.

Year	Absolute Intoxication Rates for CNS Drugs ^b^ (Number of Registered Intoxication Cases)	Number of All Registered Intoxication Cases	Relative Intoxication Rates for CNS Drugs ^b^ (95% CI) ^a^
2015	219	824	0.266 (0.232–0.300)
2016	314	983	0.319 (0.289–0.348)
2017	338	888	0.381 (0.349–0.413)
2018	353	1081	0.327 (0.299–0.355)
2019	471	1445	0.326 (0.302–0.350)
2020	445	1203	0.370 (0.342–0.397)
2021	500	1397	0.358 (0.332–0.383)
2015–2021	2640	7821	0.338 (0.328–0.348)

^a^ CI: confidence interval; ^b^ CNS: central nervous system.

**Table 3 pharmaceuticals-17-00143-t003:** Intoxication rates and intoxication/prescribing ratio values for psychotropic drug groups, 7 year period (2015–2021).

Psychotropic Drug Group	2015–2021 Period			
	Number of Intoxication Cases	Relative Intoxication Rates	7 Year IPR ^a^	Common Signs and Symptoms of Intoxication ^b^
anxiolytics	1075	0.138 (0.130–0.145)	1.42 (1.34–1.50)	somnolence (57%), coma (17%), sedation (12%), hypotension (8%), tachycardia (6%), nausea/vomiting (6%), dizziness (5%), respiratory depression (4%), agitation/aggression (4%), sensory disturbances (4%)
antipsychotics	806	0.103 (0.096–0.110)	1.07 (1.00–1.14)	somnolence (40%), tachycardia (30%), agitation/aggression (15%), sedation (10%), coma (10%), (orthostatic) hypotension (9%), nausea/vomiting (9%), dizziness (8%), sensory disturbances (8%), dysarthria (6%), QTc prolongation (5%), confusion (5%), miosis (5%)
hypnotics/ sedatives	523	0.067 (0.061–0.072)	0.87 (0.79–0.94)	somnolence (54%), coma (13%), sedation (12%), tachycardia (10%), agitation/aggression (5%), sensory disturbances (4%), hypotension (4%), confusion (4%), respiratory depression (4%), nausea/vomiting (4%), bradycardia (4%),
antiepileptics	361	0.046 (0.042–0.051)	0.52 (0.47–0.58)	somnolence (31%), coma (15%), sedation (15%), tachycardia (15%), dizziness (13%), nausea/vomiting (11%), agitation/aggression (11%), seizures (9%), respiratory depression (8%), hyperammonemia (8%), restlessness (6%), increased serum lactate levels (5%), confusion (5%), hypotension (5%), ataxia (4%), metabolic acidosis (4%), dysarthria (4%), sensory disturbances (4%)
antidepressants	576	0.074 (0.068–0.080)	0.16 (0.15–0.17)	nausea/vomiting (23%), tachycardia (21%), somnolence (17%), agitation/aggression (12%), sedation (8%), dizziness (8%), tremor (6%), QTc prolongation (6%), seizures (5%), coma (4%), sensory disturbances (4%), hypertension (4%)

^a^ IPR: intoxication/prescribing ratio; ^b^ most frequently registered signs and symptoms for mono-ingestions.

**Table 4 pharmaceuticals-17-00143-t004:** Intoxication rates and intoxication/prescribing ratio values for individual psychotropic drugs, 7 year period (2015–2021).

Psychotropic Drug	Number of Intoxication Cases	Relative Intoxication Rates	7 Year IPR ^a^
midazolam	88	0.011 (0.009–0.014)	26.59 (21.06–32.11)
chlormethiazole	75	0.010 (0.007–0.012)	8.09 (6.27–9.91)
clonazepam	107	0.014 (0.011–0.016)	6.08 (4.93–7.22)
sulpiride	46	0.006 (0.004–0.008)	3.51 (2.50–4.52)
quetiapine	459	0.059 (0.053–0.064)	2.49 (2.27–2.72)
promazine	35	0.004 (0.003–0.006)	2.16 (1.45–2.88)
diazepam	284	0.036 (0.032–0.040)	1.95 (1.73–2.17)
alprazolam	398	0.051 (0.046–0.056)	1.45 (1.31–1.59)
clozapine	58	0.007 (0.006–0.009)	1.19 (0.89–1.50)
bromazepam	167	0.021 (0.018–0.025)	1.17 (1.00–1.35)
nitrazepam	31	0.004 (0.003–0.005)	1.02 (0.66–1.37)
risperidone	74	0.009 (0.007–0.012)	0.90 (0.70–1.11)
lamotrigine	76	0.010 (0.008–0.012)	0.87 (0.68–1.07)
lorazepam	139	0.018 (0.015–0.021)	0.85 (0.71–0.99)
trazodone	58	0.007 (0.006–0.009)	0.72 (0.54–0.91)
haloperidol	32	0.004 (0.003–0.006)	0.72 (0.47–0.96)
bupropion	47	0.006 (0.004–0.008)	0.67 (0.48–0.86)
olanzapine	120	0.015 (0.013–0.018)	0.63 (0.51–0.74)
valproic acid	54	0.007 (0.005–0.009)	0.60 (0.44–0.76)
zolpidem	323	0.041 (0.037–0.046)	0.60 (0.53–0.66)
pregabalin	90	0.012 (0.009–0.014)	0.43 (0.34–0.51)
aripiprazole	29	0.004 (0.002–0.005)	0.42 (0.27–0.57)
mirtazapine	94	0.012 (0.010–0.014)	0.34 (0.27–0.41)
carbamazepine	23	0.003 (0.002–0.004)	0.30 (0.18–0.43)
paroxetine	64	0.008 (0.006–0.010)	0.20 (0.15–0.25)
sertraline	147	0.019 (0.016–0.022)	0.15 (0.13–0.18)
venlafaxine	39	0.005 (0.003–0.007)	0.15 (0.10–0.20)
duloxetine	42	0.005 (0.004–0.007)	0.11 (0.08–0.14)
escitalopram	101	0.013 (0.010–0.015)	0.11 (0.09–0.13)

^a^ IPR: intoxication/prescribing ratio.

**Table 5 pharmaceuticals-17-00143-t005:** Correlation between the trends in prescribing and intoxication rates for psychotropic drugs.

Psychotropic Drugs	Time Correlation					
	Prescribing Rates of the Year of Intoxication Events		Prescribing Rates 1 Year before the Intoxication Events		Prescribing Rates 2 Years before the Intoxication Events	
	R ^a^	*p*-Value	R ^a^	*p*-Value	R ^a^	*p*-Value
**antipsychotics**	0.767	0.044 *	0.691	0.085	0.766	0.045 *
olanzapine	0.743	0.056	0.803	0.030 *	0.871	0.011 *
quetiapine	0.802	0.030 *	0.781	0.038 *	0.820	0.024 *
**antidepressants**	0.707	0.075	0.810	0.027 *	0.828	0.021 *
sertraline	0.806	0.029 *	0.821	0.024 *	0.874	0.010 *
escitalopram	0.122	0.794	0.542	0.209	0.676	0.095
mirtazapine	0.907	0.005 *	0.877	0.009 *	0.865	0.012 *
**antiepileptics**	0.963	<0.001 *	0.949	0.001 *	0.946	0.001 *
clonazepam	0.775	0.041 *	0.756	0.049 *	0.728	0.064
pregabalin	0.950	0.001 *	0.960	0.001 *	0.954	0.001 *
**anxiolytics**	−0.642	0.120	−0.573	0.179	−0.522	0.230
diazepam	−0.123	0.793	−0.563	0.188	0.096	0.837
lorazepam	−0.647	0.116	−0.880	0.009 **	−0.768	0.044 **
bromazepam	−0.878	0.009 **	−0.765	0.045 **	−0.878	0.009 **
alprazolam	−0.358	0.431	−0.106	0.820	0.048	0.918
**hypnotics/sedatives**	−0.470	0.287	−0.403	0.370	−0.288	0.532
zolpidem	−0.337	0.460	−0.283	0.538	−0.166	0.723
midazolam	−0.140	0.764	0.008	0.987	0.068	0.885

^a^ Pearson coefficient; * statistical significance; ** negative correlation.

## Data Availability

Data on correlation analysis are provided in a Appendix A. Other data are available upon request from the authors.

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
