# Peer review of "Association between Prescribing and Intoxication Rates for Selected Psychotropic Drugs: A Longitudinal Observational Study"

_pharmaceuticals, 2024, doi:10.3390/ph17010143_

Round 1
Reviewer 1 Report
Comments and Suggestions for Authors
In this article by Matej Dobravc Verbic and colleagues, the authors have performed an accurate analysis of the association between prescribing and intoxication rates for different psychotropic derugs. The article could have a potential interest into the toxic-epidemiology field and shows that anxiolytics and antipsychotics are the most frequently implicated groups.
The analysis is well conducted and the statistical methodology seems to be appropriate. However, I have few suggestions that in my oipinion could extend the interest toward this article.
1) The authors generally talk about intoxication events therefore I was wondering whether they have recorded the type of these adverse events. In my opinion, in order to add also a clinical interest a table summarizing the most recurrent symptoms together with the intoxication rate for each drug classes could be useful for clinicians during ther diagnosis process for performing an accurate identification of toxicity signs and the drugs responsible for them.
2) Considering nowdays the spread of psychotropic drugs' prescription among children and adolescents, it could be useful in terms of pediatric epidemiology whether the authors could consider also the intentional and unintentional intoxication events (with or without prescription) that involve adolescents.
Comments on the Quality of English Language
The overall English language is good however a moderate revision is required. For example, "prescription psychotopic drugs" is recurrent troughout the manuscript. In my opinion it should be "prescription of psychotropic drugs".
Author Response
Thank you very much for taking the time to review this manuscript. While it was not possible to include the adolescent group, we presented the common signs and symptoms of intoxication as suggested. Please find the detailed responses below and the corresponding revisions/corrections highlighted/in track changes in the re-submitted files.
Best regards,
Matej Dobravc Verbič
1) We agree with this comment. We have extracted the additional data regarding the clinical picture of intoxications. We added common signs and symptoms of intoxication in the table for psychotropic drug groups. Signs and symptoms are stated for the cases of mono-ingestions only (in other cases, the clinical picture may be misleading and related to other agents involved).
2) Unfortunately, our chosen focus group were adult patients and we have gathered the data exclusively for the adults, both regarding the intoxications as well as prescribing rates. Additional data for the adolescents cannot be obtained from the databases we have gained access to. Therefore, it was not possible to include the adolescent population in the manuscript.
Response to Comments on the Quality of English Language:
We followed the advice and performed a revision of English language using InstaTEXT and Writefull.
“Prescription drugs” is a common term, which we used in the text to emphasise that other psychotropic drugs (e.g. drugs of abuse) were not included in the study. We have now changed the word order in this phrase into “psychotropic prescription drugs”.
Reviewer 2 Report
Comments and Suggestions for Authors
The manuscript is interesting. However, some issues should be clarified.
1. Did Authors take into the consideration that COVID-19 pandemic have also the impact on the intoxication rate or overdose?
2. Did Authors take into consideration sex of the patient as a variable? The intoxiation rate could be different in males and females.
3. Were there any inclusion/exclusion criteria in the study?
Author Response
Thank you very much for taking the time to review this manuscript. We appreciated the recommendations. Please find the detailed responses below and the corresponding revisions/corrections highlighted/in track changes in the re-submitted files.
Best regards,
Matej Dobravc Verbič
1) We considered COVID19, but we have not put this in the previous version of the manuscript. We have now added a paragraph discussing COVID-19. However, we did not recognise an influence of the pandemic on our study results.
2) No, stratification by sex (and age) was not the purpose of the study. Sex was not considered as a variable. Unfortunately, the data on prescribing stratified by sex were not available to us, so it was not possible to determine the association of intoxication and prescribing rates in men and women separately. However, we have now added the information on sex (and median age) for the intoxication events in the Results. The intoxication rates were slightly higher in women. We have added a study limitation in the revised version, explaining that our data were not stratified by age, gender or disease.
3) Yes, inclusion and exclusion criteria have been described in the initial version of the manuscript in the Methods section (inclusion criteria: adult population [≥18 years old], intentional or unintentional intoxication with a psychotropic prescription drug; exclusion criteria: <18 years old; intoxication not involving a psychotropic prescription drug). We have made some clarifications regarding the criteria in the Methods section of the revised version to avoid any doubt: intoxication cases were included regardless of the severity, outcome and potential co-ingestants.
Reviewer 3 Report
Comments and Suggestions for Authors
Thank you for your submission.
The abstract does not contain adequate methodological details e.g. centre/country, source of data for poisonings and prescribing.
This statement is incorrect: “However, no direct quantitative comparison of the risk for intoxication with psychotropic drugs from different groups in relation to their prescribing rates has been performed”. Amongst other examples are:
· Shah A, Zhinchin G, Zarate-Escudero S, et al. The relationship between the prescription of psychotropic drugs and suicide rates in older people in England and Wales. Int J Soc Psychiatry 2014;60(1):83-8.
· Shah A, Lodhi L. The impact of trends in psychotropic prescribing on the method of suicide in the elderly. Med Sci Law 2005;45(2):115-20.
Are the “prescribing” rates reported actually for prescribing or dispensing? In fact, the manuscript states that “Prescribing rates were determined from the dispensing data...” Either the term “dispensing data” should be used throughout the manuscript or the methods and limitations should clearly state the prescribing can only be estimated (not determined) from dispensing data.
Is the dispensing data (DDDs) only for adults or the whole population? How was this calculated?
Are some of the drugs listed in Table 4 used extensively in hospitals? (e.g. midazolam, chlormethiazole). Did the data sources (especially for dispensing) include hospital data?
The strong negative correlations in Table are not adequately or logically explained. Why would the imposition of prescribing restrictions increase the intoxication rates relative to prescribing?
The discussion is very long and could be shortened.
Comments on the Quality of English LanguageEnglish is fine.
Author Response
Thank you very much for taking the time to review this manuscript. We appreciated the constructive comments and have considered all of the recommendations. Please find the detailed responses below and the corresponding revisions/corrections highlighted/in track changes in the re-submitted files.
Best regards,
Matej Dobravc Verbic
- The abstract does not contain adequate methodological details e.g. centre/country, source of data for poisonings and prescribing. These details are now included in the abstract. Some additional changes had to be made in the abstract to not exceed the required limitation of 200 words.
- This statement is incorrect: “However, no direct quantitative comparison of the risk for intoxication with psychotropic drugs from different groups in relation to their prescribing rates has been performed”. Amongst other examples are: Shah A, Zhinchin G, Zarate-Escudero S, et al. The relationship between the prescription of psychotropic drugs and suicide rates in older people in England and Wales. Int J Soc Psychiatry 2014;60(1):83-8. Shah A, Lodhi L. The impact of trends in psychotropic prescribing on the method of suicide in the elderly. Med Sci Law 2005;45(2):115-20. Thank you for the recommended articles. It is true that relationship between prescribing and suicide rates (due to non-specific reasons, or due to overdoses using the drugs prescribed) has been assessed in previous research, which we have now included in the Introduction. However, suicide or overdose rates do not give the same information as intoxication rates in our study, which covered any (intentional or non-intentional) ingestion of the amount that exceeds the therapeutic threshold, regardless of the severity of the intoxication. We found no previously published work that would offer a similar quantitative comparison of intoxication rates in relation to the prescribing. We have added an explanation in the Methods section, that intoxication events were included regardless of the severity of the symptoms and outcome. “The risk for intoxication” in the statement mentioned above is changed in “the overall risk for intoxication” in the new version of the manuscript.
- Are the “prescribing” rates reported actually for prescribing or dispensing? In fact, the manuscript states that “Prescribing rates were determined from the dispensing data...” Either the term “dispensing data” should be used throughout the manuscript or the methods and limitations should clearly state the prescribing can only be estimated (not determined) from dispensing data. The reported rates are the dispensing rates of the outpatient prescriptions, which is now stated also in the Abstract and the Limitations sections. In Methods section, we have changed the word “determined” into “estimated” as recommended.
- Is the dispensing data (DDDs) only for adults or the whole population? How was this calculated? The dispensing data were obtained upon request from the Health Insurance Institute of Slovenia exclusively for adult population. Prescriptions were included if the patient was ≥18 years old when the prescription was dispensed. Number of DDDs was calculated by the Health Insurance Institute of Slovenia. The calculations were based on internationally defined information for every drug (i.e. available at this link: https://www.whocc.no/atc_ddd_index/ ). DDDs for adults for oral route of administration were used for calculation. “Oral route of administration” is now stated in the Methods section to avoid any doubt.
- Are some of the drugs listed in Table 4 used extensively in hospitals? (e.g. midazolam, chlormethiazole). Yes, but this did not influence our study results because only the outpatient prescriptions were included.
- Did the data sources (especially for dispensing) include hospital data? No, the data regarding prescribing/dispensing were specifically for outpatient prescriptions. We have now emphasised in the Methods section that the dispensing data were for outpatient prescriptions.
- The strong negative correlations in Table are not adequately or logically explained. Why would the imposition of prescribing restrictions increase the intoxication rates relative to prescribing? The prescribing restrictions may have been a confounding variable. While the rates of covered prescriptions decreased, the rates of paid prescriptions increased, but we have verified that these had little or no impact on the correlation analysis. More importantly, restricted prescribing may have encouraged patients to obtain the drugs via a different route, e.g. from neighbour countries, where it is possible to buy these drugs without prescription. It was possible for us to identify this problem for midazolam, because it is sold under a different brand name. Some other benzodiazepines (e.g. bromazepam and lorazepam – the two drugs with a significant negative correlation and with restricted prescribing) are registered abroad under the same brand name as in Slovenia. In the intoxication setting, it is difficult to distinguish, where the drug originates from, and the true origin of the drug is unlikely to have been registered. Unfortunately, we have no evidence to confirm this assumption. Additionally, we have included the small sample size for individual drugs as a possible confounding factor in the discussion, and also have included it in the Study limitations section (together with relatively short [7-year] study period and multiple comparisons performed in the correlation analysis which may increase the likelihood of type-I error).
- The discussion is very long and could be shortened. The discussion has been shortened where possible to do so without losing relevant information. As recommended by other reviewers, we added a paragraph discussing the impact of Covid-19.
Round 2
Reviewer 3 Report
Comments and Suggestions for Authors
Thank you for the revision.
Comments on the Quality of English LanguageMainly fine.